# Induced Periosteum-Mimicking Membrane with Cell Barrier and Multipotential Stromal Cell (MSC) Homing Functionalities

**DOI:** 10.3390/ijms21155233

**Published:** 2020-07-23

**Authors:** Heather E. Owston, Katrina M. Moisley, Giuseppe Tronci, Stephen J. Russell, Peter V. Giannoudis, Elena Jones

**Affiliations:** 1Clothworkers’ Centre for Textile Materials Innovation for Healthcare, School of Design, University of Leeds, Leeds LS2 9JT, UK; g.tronci@leeds.ac.uk (G.T.); s.j.russell@leeds.ac.uk (S.J.R.); 2Leeds Institute of Rheumatic and Musculoskeletal Medicine, University of Leeds, Leeds LS9 7TF, UK; Katrina.Moisley@electrospinning.co.uk (K.M.M.); pgiannoudi@aol.com (P.V.G.); e.jones@leeds.ac.uk (E.J.); 3Institute of Medical and Biological Engineering, University of Leeds, Leeds LS2 9JT, UK; 4School of Dentistry, St. James’s University Hospital, University of Leeds, Leeds LS9 7TF, UK; 5Academic Department of Trauma & Orthopaedic Surgery, Leeds General Infirmary, Leeds LS2 9NS, UK

**Keywords:** periosteum, induced membrane, barrier membrane, free surface electrospinning, nonwovens

## Abstract

The current management of critical size bone defects (CSBDs) remains challenging and requires multiple surgeries. To reduce the number of surgeries, wrapping a biodegradable fibrous membrane around the defect to contain the graft and carry biological stimulants for repair is highly desirable. Poly(ε-caprolactone) (PCL) can be utilised to realise nonwoven fibrous barrier-like structures through free surface electrospinning (FSE). Human periosteum and induced membrane (IM) samples informed the development of an FSE membrane to support platelet lysate (PL) absorption, multipotential stromal cells (MSC) growth, and the prevention of cell migration. Although thinner than IM, periosteum presented a more mature vascular system with a significantly larger blood vessel diameter. The electrospun membrane (PCL3%-E) exhibited randomly configured nanoscale fibres that were successfully customised to introduce pores of increased diameter, without compromising tensile properties. Additional to the PL absorption and release capabilities needed for MSC attraction and growth, PCL3%-E also provided a favourable surface for the proliferation and alignment of periosteum- and bone marrow derived-MSCs, whilst possessing a barrier function to cell migration. These results demonstrate the development of a promising biodegradable barrier membrane enabling PL release and MSC colonisation, two key functionalities needed for the in situ formation of a transitional periosteum-like structure, enabling movement towards single-surgery CSBD reconstruction.

## 1. Introduction

Critical size bone defects (CSBD) and fracture non-union pose a problem for patients, orthopaedic surgeons, and healthcare services worldwide. CSBDs are classified as greater than 1–2 cm in length or >50% of the bone circumference [1] and form through bone loss following trauma, tumour resection, infection, or established non-union [2]. Non-union of fractures refer to those that fail to heal within nine months and represent 2–10% of all fractures [3,4,5]. Current treatment strategies for non-united fractures are complicated and costly (approximately £29,000 per patient) [6,7], whereby patients often require multiple surgeries. The likelihood of a successful union depends on the extent of bone or periosteum (outermost layer of bone) loss, location, presence of infection, soft tissue envelope condition, provision of adequate mechanical stability, as well as patient related factors, including age and co-morbidities [8].

A classic clinical approach to surgical repair includes bone debridement from the defect site, creating an aseptic vascular environment to then be filled with autograft material, typically from the iliac crest. More recent approaches also include autologous sources of growth factors like platelet lysate (PL) or platelet rich plasma (PRP) and multipotential stromal cells (MSC), with osteogenic and chondrogenic differentiation potential, vital for bone fracture healing [9,10,11]. In the case of known or suspected infection, the two-step Masquelet technique is often carried out, following debridement the defect is filled with antibiotic-loaded bone cement (polymethyl methacrylate (PMMA)). A second procedure is carried out after 6–12 weeks, allowing for the bone infection to be cleared and an ‘induced periosteum’ or ‘induced membrane’ (IM) forms around the PMMA. Following PMMA removal, the IM creates a biological chamber to contain grafting material as well as stimulating osteogenesis [12]. However, whilst the Masquelet technique has a union rate of over 80% [13,14], IM formation relies on the presence of PMMA and requires two separate surgeries, which can be costly and inconvenient for the patient.

Surgical technique improvements are needed and the implantation of a biocompatible membrane has been proposed. If wrapped around a CSBD during surgery, the membrane could act as a barrier and cell containment device [15]. By degrading over time, it would allow for cell homing and supporting periosteal regrowth, as well as preventing soft tissue invasion into the defect site, a known impediment to bone healing [16]. Commercially available membranes are mainly employed for maxillofacial or oral applications, however, the majority are collagen based, and thus display rapid degradation times in vivo, with complete degradation observed within 1–9 weeks [17,18]. In contrast, orthopaedic applications would require membrane stability in vivo for 12 weeks, as the minimal time thought to achieve bone healing [15,19,20], whilst displaying macroscopic elasticity to enable conformation around the defect site. In addition, current products do not mimic aspects of the periosteum or IM, key to ensuring successful bone reconstruction.

Periosteum lines the external surface of bone, made from two layers, namely the outer fibrous layer formed mainly of collagen fibres, and particularly vascular acting as a barrier, controlling fluid flow in and out of the bone [21]. The inner cambium layer lines the bone and is highly cellular, containing osteoblasts, osteoblastic progenitors, and MSCs, making this tissue important to fracture healing [22]. In response to fracture, the periosteum thickens, cell proliferation increases, and periosteal derived cells mobilise and migrate into the fracture callus. How this effects successful bone union is apparent following periosteal stripping, which results in reduced or delayed fracture healing [15,23]. The IM is often referred to as ‘induced periosteum’ due to its similar anatomical location, bilayer structure, and high collagen content, however its microarchitecture is considerably denser and significantly thicker than periosteum [24].

Aiming towards single-step, patient-friendly surgical approaches, human periosteum and IM could serve as a template to guide the design of biodegradable membranes to stimulate endogenous bone regeneration. The membrane should be designed (i) to act as a barrier to cell infiltration, (ii) enable loading of bioactive osteoinductive supplements during orthopaedic surgery, such as bone marrow aspirate (BMA) or PL, and (iii) to degrade in a controlled fashion in vivo. Herein, we developed a barrier membrane for the treatment of CSBDs, by firstly characterising the architecture of human periosteum and IM. Our design strategy was to select a clinically approved biodegradable material, i.e., poly(ε-caprolactone) (PCL), given its extensive body of literature, prevalent use in existing medical devices, and an established safety and functionality understanding [25,26,27,28]. Then, to overcome the well-known production limitations of needle electrospinning [29,30] and enable scalable manufacturing consistent with industrial needs, free surface electrospinning (FSE) was employed to a membrane suitable for application in CSBDs. Together with the physical characterisation, functional testing of the membrane was carried out in vitro to assess the absorption and release of PL, MSCs interactions, as well as the barrier functionality to cell migration.

## 2. Results

### 2.1. Architecture of Human Periosteum and Induced Membrane

Human periosteum (*n* = 6) and IM (*n* = 6) samples were obtained from patients undergoing orthopaedic surgery. The mean donor age was 52 ± 22 years (Periosteum; range: 23–80 years) and 43 ± 10 years (IM, range: 32–58 years) and samples were harvested mainly from the femur, but also humerus, iliac crest, tibia, and ulna (Appendix A). Both tissue types were highly vascularised and cellularised (Figure 1A,D), with evidence of IM having a more distinct bilayer architecture, where a thin layer, in contact with the PMMA cement in situ, appeared more fibrous (Figure 1D). With periosteal samples, the bilayer structure was much less distinct and often not present (Figure 1A). However, for both tissues, collagen fibres were shown to be the main structural component, as seen through Picro Sirius Red (PSR) staining (Figure 1B,E). When quantified, IM samples were significantly thicker (1.94 ± 0.22 mm) than periosteum (1.17 ± 0.50 mm, unpaired *t*-test, *p* = 0.003) (Figure 1G). IM also contained blood vessels with a significantly smaller diameter (15.38 ± 4.38 μm) compared to periosteum (22.74 ± 2.19 μm, unpaired *t*-test, *p* = 0.004) (Figure 1H). In addition, the density of blood vessels was higher for periosteum (14.7 ± 8.4 blood vessels per mm^2^ (BV/mm^2^)) compared to IM (8.8 ± 2.2 BV/mm^2^), however this difference was not significant, potentially due to variation in the periosteum samples (Figure 1I).

### 2.2. Free Surface Electrospun Membrane Characterisation

A membrane was electrospun via FSE (PCL3%-E) using a 3% *w*/*v* polymer solution to produce an easy to handle material compiled from randomly orientated nanoscale fibres (Figure 2A,B), with a mean fibre diameter of 354 ± 174 nm (range: 116–788 nm). This interconnected fibre network displayed a porous structure, which when quantified, mean pore size was 1.37 ± 0.06 μm, ranging from 1 to 35 μm (Figure 2F). However, pores over 2 μm were rare, accounting for 2.61 ± 0.05% of all pores.

An integral aspect of the membrane design was wetting and liquid absorbance, hence the WCA was measured for PCL3%-E in comparison to a 3% PCL cast film (PCL3%-Film). The PCL3%-Film was hydrophobic, showing a WCA of 125 ± 1°, maintained over 15 s (Figure 2G). In comparison, the fibrous PCL3%-E membranes had a smaller initial WCA of 64 ± 1°, which was reduced to 15 ± 4° after eight seconds, following which WCA was undetectable.

Post manufacture, folding, and heat bonding of PCL3%-E enabled an increase in the membrane thickness from 97 ± 13 μm to 408 ± 43 μm. Additionally, larger pores (diameter: 494 ± 41 μm) were introduced using a laser cutter (Figure 2C–E), so that the resulting membrane could act as a physical barrier whilst allowing nutrient exchange upon implantation. The larger pores introduced risked negatively affecting the mechanical properties of the membrane, so tensile tests were evaluated for PCL3%-E and laser cut samples (PCL3%-S and PCL3%-D) (Figure 2C,D). Interestingly, the presence of sparsely distributed pores (4 mm gap, PCL3%-S) in the membrane resulted in no significant difference (one-way ANOVA, *p* < 0.05) in the ultimate tensile strength (UTS) (PCL3%-E—1.0 ± 0.2 MPa, PCL3%-S—1.4 ± 0.3 MPa), strain at break (PCL3%-E—54 ± 5%, PCL3%-S—50 ± 8%), and elastic modulus (PCL3%-E—4.4 ± 0.8 MPa, PCL3%-S—4.7 ± 0.7 MPa) (Figure 2H). Mechanical properties were however significantly affected (one-way ANOVA, *p* < 0.05) when the inter-pore distance was reduced to 2 mm (PCL3%-D), with significantly detrimental effects on UTS (0.5 ± 0.1 MPa), strain at break (37 ± 4%), and elastic modulus (2.1 ± 0.3 MPa). This was likely due to increased stress concentration given the higher open area (Figure 2H).

### 2.3. PL Absorption and Release Profile of PCL3%-E

To quantify the PL loading/release capability of PCL3%-E and a commercially available control scaffold, Mucograft^®^, were incubated in vitro with PL solutions and the absorption volume (per unit area) was quantified. The absorption volume indicated that PCL3%-E had a similar absorption capacity to Mucograft^®^ (Figure 3A). This promising finding demonstrated that PCL3%-E displays at least the same loading efficiency as a commercial equivalent, although limited replicates prevent strong conclusions being made. Following PL uptake, the release profile of the absorbed proteins was studied. Following 1 h incubation, the protein release was lower for PCL3%-E (Figure 3B). Further detail of the cumulative release profiles for PCL3%-E over 60 h indicated release variation between sample replicates, although the majority of PL release occurred within the first 2 h of incubation (Figure 3C). Closer analysis of the release profile in the first 2 h showed that despite variation between the PCL3%-E samples, Mucograft^®^ released less protein, and more gradually (Figure 3D).

### 2.4. Cellular Interactions with PCL3%-E

In addition to being a carrier for autologous biological healing stimulants, a membrane for CSBDs should also support cellular attachment and proliferation, in particular that of periosteal cells to support periosteal regrowth during bone regeneration. To assess this functionality in vitro, PCL3%-E was cultured with MSCs derived from either human BM or periosteum, at either low (10^3^ cells/sample) or high (10^4^ cells/sample) seeding densities and grown for up to 28 days.

Both MSC types attached to PCL3%-E at similar rates following four days of culture, regardless of seeding density (Figure 4A,D,G,J). At the low seeding density (10^3^ cells/sample), the proliferation rates of both BM and periosteum-derived MSCs were similar for up to 14 days. Following 28-day cell culture, the MSC homing capability of PCL3%-E was confirmed, despite differing proliferation rates being detected for both MSC types. BM MSCs had significantly increased cell numbers in comparison to periosteum MSCs at the same culture time point, whilst displaying a more spread out and spindle shaped phenotype (Figure 4A,F,M).

In contrast, at the higher density (10^4^ cells/sample), periosteum MSCs proliferated at a significantly higher rate from day 7 onwards, which continued up to 28 days and appeared to become aligned over time (Figure 4G,L,N). The angle of individual nuclei was quantified and was normally distributed, therefore full width at half maximum (FWHM) values were quantified as the spread of distribution and nuclei alignment. FWHM reduced slightly over 28 days. Therefore, nuclei alignment was not confirmed within this time frame for both experimental conditions with BM MSCs. A stark reduction in FWHM and consequent nuclei alignment was seen for periosteum MSCs at both seeding densities, with a significant reduction detected from day 7 onwards in comparison to values recorded at day 4 (Figure 4O). Thus, periosteum MSCs aligned with ease at both seeding densities, while BM MSCs took longer to respond at lower seeding densities, showing preference to align at higher seeding densities. This suggests the production of a material that will support periosteal regrowth.

Another key requirement to ensure suitability for CSBDs is for PCL3%-E to act as a physical barrier, preventing soft tissue ingrowth from the surrounding musculature. In order to functionally test this capability a ‘modified transwell assay’ was developed. PCL3%-E and PCL3%-S (Figure 2C) were trapped within a MINUSHEET^®^ tissue carrier and MINUSHEET^®^ tension ring, floating on serum containing culture media. MSCs, serum starved for 24 h, suspended in serum free culture media were seeded on top of the construct, creating a serum gradient across PCL3%-E to drive MSC migration through the membrane (Figure 5). Cell migration through PCL3%-E was compared to PCL3%-S over a one-week culture period, and the constructs were imaged on both sides.

Clear cellular attachment and proliferation was seen over a week when imaged from the top surface for both PCL3%-E and PCL3%-S, however to a lesser extent for the latter (Figure 5). It was noted that there was lower cellular attachment in the immediate vicinity of the laser cut holes (Figure 5D-F). When imaged from the bottom surface to visualise any cells that had migrated through either construct, neither showed evidence of cellular material (Figure 5, inserts).

## 3. Discussion

In this work, we developed a biodegradable induced periosteum-like membrane via FSE and then tested its future clinical functionalities, i.e., as a carrier of PL, MSC homing, and preventing cellular infiltration. Both the native periosteum and IM have key roles during uneventful bone regeneration [12]. In an effort to improve the surgical repair of CSBD and reduce the number of surgeries needed per patient, the development of a membrane that replaces the role of periosteum and IM is needed. Clinically, the membrane would be wrapped around the defect site, acting as a containment factor to enclose the grafting material that is remodeled overtime to support fracture union. Having a physical barrier can also prevent muscular or soft tissue ingrowth into the defect site, as well as preventing fibroblastic infiltration which can hamper osteogenic repopulation of the defect site, impeding bone regeneration [16,31,32].

Histological analysis of human periosteum and IM was an important step to inform the membrane development. In agreement with our previous work, human IM was approximately 1.7-fold thicker than periosteum [24], however this differs from the 20-fold increase in rats [33], highlighting clear species differences and the importance of studying human tissue. Additionally, thickness variation in both tissues was influenced by donor age or anatomical location, suggestive that when developing a new membrane mimicking this precisely is not necessary. In addition, although at a similar density, blood vessels within periosteum were significantly larger than IM. This was in line with previous studies and expected due to the developing nature of IM in comparison to the established resident periosteum, but presents novel quantification in human samples [24,33]. Interestingly, linear increases to IM thickness and blood vessel density with time was recorded in rats, that decreased following graft material placement [34].

FSE was utilised to overcome the production scale limitations of needle-based electrospinning [30] and created a highly porous nonwoven membrane constructed of nanoscale diameter fibres. More than 95% of pores in PCL3%-E had a diameter of 1–2 μm, such that MSCs and fibroblasts are unlikely to penetrate due to size exclusion as BM MSCs have a diameter of 12–21 μm [35,36,37]. Therefore, the membrane was expected to act as a true containment device, preventing fibroblastic infiltration and MSC loss from the defect, whilst allowing for the transport of nutrients or other solutes through the membrane. As a proof-of principle for post-manufacturing customisation, laser cut pores with increased diameter were introduced with a sparse inter-pore distance (4 mm gap, PCL3%-S), that did not negatively influence tensile strength. The relationship between increased pore frequency or size and a reduction in material strength is known, demonstrating that a balance needs to be met between the two [38]. There was similar post-manufacturing customisation of scaffolds to regulate transparency for corneal applications [39], but this remains novel to the orthopaedic field. As human periosteum or IM thickness is ~1 mm, increased membrane thickness could be achieved if desired by a thermal bonding of multiple layers of PCL3%-E, allowing for increased absorption. Once implanted, it would be expected that membrane thickness and pore size would decrease and increase, respectively, over time according to a bulk hydrolytic degradation mechanism [40].

Currently, autologous sources of BMA, BMA concentrate, PL or PRP are mixed with bone grafting material or injected into the defect [41,42]. If injected directly in situ, this can be lost quickly from the intended delivery site, potentially negating its use. Absorption onto the membrane, prior to implantation, would allow for the defect site to be surrounded with a slower release of growth factors (Figure 6). Despite PCL being a hydrophobic synthetic polymer [26], it exhibited prompt water absorbency, similar to needle electrospun membranes [43]. Parity between PCL3%-E and Mucograft^®^, a commercially available collagen membrane used in oral soft tissue regeneration, was shown for the absorption and release of commercially available PL (a standardised and consistent product). A recent evaluation of Mucograft has shown a gradual, but high early release of protein within the first six hours [44]. Importantly for future clinical translation, the release of PL from PCL3%-E into the surrounding media also supports cellular attachment and proliferation of nearby MSCs [45]. With a single replicate for Mucograft^®^, the conclusions that can be taken from these experiments are somewhat limited, however as a commercial product, Mucograft^®^ was expected to show limited batch variation.

Once implanted into a defect site the membrane should support periosteal regrowth or MSC migration and expansion (from periosteum, bone marrow cavity and endosteum) as the first steps in the formation of native tissues’ structures during bone regeneration [46]. The membrane will also be in direct contact with the bone graft and bone filler material (either synthetic or natural) that is implanted into the defect site, and thus MSCs already contained within the filler material. As previously mentioned, BMA is routinely mixed with bone grafting material during CSBD repair surgeries. However, periosteum, a known ‘reservoir’ of MSCs, is overlooked as an MSC source [47,48,49] (Figure 6). As periosteum and BM MSCs are key players for bone regeneration [15,23], interactions with PCL3%-E were robustly tested, showing differing behaviours. For the first time, morphological differences in human donor matched MSCs from BM and periosteum has been shown when seeded onto a periosteal substitute membrane. No differences were seen over three days for rat derived periosteum and BM MSCs grown onto cellulose membranes [50]. Here, changes in the morphology and proliferation were seen after a week, showing the importance of monitoring long term cellular growth.

Previously, cellular alignment was thought to be driven by fibre alignment or microgrooves within the underlying material [51,52]. Unexpectedly, both MSC types aligned over time, despite random fibre configuration. This was of particular interest as cells within periosteum align along the longitudinal axis of bone, following or guiding collagen fibre alignment [52,53], in contrast to BM MSCs, that line the surface of bone [54]. Thus, the alignment of both periosteal and BM MSCs is a critical advantage as preferential alignment of fibres compromises the isotrophy of membrane tensile strength [55]. Periosteal MSCs aligned irrespective of cell density, although at a slightly slower rate with lower seeding density, in contrast, BM MSCs preferred alignment with higher seeding density. This could be related to BM MCSs tendency to promote the formation of less frequent but larger colonies, compared to periosteum, suggestive of a preference for higher seeding densities [47]. Once implanted, PCL3%-E may likely be first populated by neighboring periosteal cells rather than BM MSCs from the fracture border. Where the alignment of cells or actin filaments on similar membranes (random fibre orientation) has been seen in the literature, it is not commented on, let alone quantified to the novel extent of this study [56,57,58,59,60]. Furthermore, seeding densities in these studies were much higher (6.9 × 10^3^ to 1.6 × 10^6^ cells per cm^2^) than in this body of work (2.5 × 10^2^ or 2.5 × 10^3^ cells per cm^2^), which aimed to mimic lower numbers of MSCs that could be realistically obtained during surgery. There is current evidence that fibre alignment also results in increased levels of osteogenic markers (gene expression and protein level) [52,56], following osteoinduction, thus future work would aim to ascertain differences in calcium deposition onto PCL3%-E for both MSC types, based on the preferential alignment of periosteum MSCs.

Current literature does not disclose testing of the barrier function of membranes *in vitro*, whilst the consensus is to merely rely on the assumption of blocking cellular infiltration through pore size [56,61]. Slightly more robust in vitro assessments have relied on histological staining and confocal imaging of attached cells with respect to the underlying membranes [57,62,63,64]. However, this has focused on cell penetration into the scaffold, not whether cells could migrate through the full membrane thickness. Therefore, a modification to the classical cell migration transwell assay was successfully developed, whereby instead of creating a serum gradient across predefined pores in a transwell insert, cell migration was driven through PCL3%-E. Over the course of a week, MSCs were unable to migrate through the thickness of PCL3%-E in addition to the laser cut pore positive control, PCL3%-S.

The future exploration of different post manufacture pore formation is needed to create smaller pores (<500 μm). Other techniques include membrane ultrasonication, known to increase pore sizes; or the use of co-spinning with a water soluble sacrificial polymer that can be ‘washed away’, leaving larger pores [65]. Alternatively, micro needles of the desired diameter could also introduce additional pores [66]. Furthermore, investigation into the natural opening of pores following in vivo degradation is needed to ascertain the changes in porosity over time. ‘Minimally manipulated’ MSC sources should be examined, where BMA, the current surgical gold standard, or periosteum micrografts, MSCs released from macerated tissue, could be combined with PCL3%-E [67,68,69]. Additionally, future in vivo studies are required to not only detail membrane degradation in vivo, but also to characterise the membranes influence on bone regeneration [70].

## 4. Materials and Methods

### 4.1. Human Tissue Collection and Ethics

Human samples of periosteum, IM, and BMA (aspirated from the iliac crest) were obtained from surgical non-union patients being treated at the Trauma Orthopaedic Unit at Leeds General Infirmary (Leeds, UK). Informed written consent was given by all patients and ethical approval was granted by the National Research Ethics Committee—Leeds East, with ethical approval number 06/Q1206/127. Periosteum (at least 5 cm away from the operating fracture site) and IM (from the center of the bone defect, 9 ± 5 weeks following PMMA cement spacer implantation) was harvested as per previous methods [24,47]. BMA was aspirated from a donor (Male, 23) through standard methods [71]. Patient characteristics of histological samples can be found in Appendix A.

### 4.2. Histology Analysis of Periosteum and Induced Membrane

PBS washed periosteum and IM were fixed in 3.75% (vol.%) formaldehyde and processed for histological assessment and stained for Haematoxylin and Eosin (H&E) and PSR (see Appendix A). Slides were digitally scanned, using the Virtual Pathology Project Service at the University of Leeds and PSR samples were imaged under polarised light [47]. Scanned images of the full tissue slice were used to quantify tissue thickness (*n* = 20 measurements), tissue surface area (drawing accurate ‘ring’ around the tissue, automatically calculating surface area), blood vessel diameter, and frequency (again drawing a ‘ring’ around the inner lumen of blood vessels). Data were exported to excel for data analysis.

### 4.3. Free Surface Electrospun Membrane Manufacture and Processing

PCL (Sigma) solutions (3% *w*/*v*) were prepared in 1,1,1,3,3,3-Hexafluoro-2-propanol (Sigma) and stirred for 24 h. Solutions were spun into fibrous nonwoven membranes (PCL3%-E) via FSE (NanoSpider Lab-200, Elmarco) on a low volume spike electrode (surface area of 0.78 cm^2^), with a spinning distance of 155 mm at 40 kV and collected onto a polypropylene spunbond fabric (Elmarco). PCL3%-E was laser-cut to create additional 200 µm diameter pores, where two distinct microarchitectures were designed, with either a sparse (PCL3%-S) or dense (PCL3%-D) inter-pore distance of 2 or 4 mm. A 100-watt CO_2_ air-cooled laser cutter, with ApS-Ethos control software created full cut-through pores using a single pass at 10% power and a velocity of 600 mm/s. For increased thickness (0.5 mm), PCL3%-E was folded by thermal annealing at 40 °C on a hot-plate and applying even pressure to the layers.

### 4.4. Physical Characterisation of the FSE Membranes

Sputter-coated (Gold, Agar Auto Sputter Coater) membranes were imaged using scanning electron microscopy (SEM) (Hitachi, S-3400N) to quantify fibre diameter (*n* = 20 measurements per image (*n* = 3)). Capillary flow porometry (Porolux 100FM) quantified pore size distribution. Tensile testing of PCL3%-E, PCL3%-S and PCL3%-D (non folded) (15 × 30 mm, *n* = 6) was carried out using a 100 N load cell (Titan), gauge length of 20 mm and a crosshead speed of 20 mm/min until breakpoint (F_max_) to create stress (σ, MPa) strain (ε, mm/mm) curves. A water contact angle (WCA) of PCL3%-E and pore-free films (n = 3) was measured, using a contact angle goniometer (CAM 200, KSV Instruments, Ltd., Gothenburg, Sweden), twice a second over 15 s by a CCD fire wire camera.

### 4.5. PCL3%-E Membrane Absorbance and Release of Platelet Lysate

Folded PCL3%-E membranes (*n* = 2) and Mucograft^®^ (Geistlich Pharma) (*n* = 1), with an area of 1 cm^2^ were dried at 40 °C for 12 h before being weighed pre- (weight_dry_) and post-soaking (weight_soaked_) in 1 mL of 100% PL (Cook Regentec) for 15 min at RT to mimic the clinical setting [10,12]. The weight increase (%) was calculated as an indicator of absorption capacity and normalised to the sample area. PL soaked PCL3%-E (*n* = 2) and Mucograft^®^ were incubated in simulated body fluid (SBF) at 37 °C, and fluid samples (60 μL) were collected sacrificially over time replaced with an equal volume of SBF. A bicinchoninic acid (BCA) assay was carried out, following the manufacturers protocol, to quantify protein concentration (μg/mL) and due to the sacrificial nature of the release assay, a dilution factor was also accounted for in each sample.

### 4.6. In Vitro MSC Attachment onto PCL3%-E

MSCs, isolated from iliac crest BMA and femoral periosteum (Male, 17) as previously described (31), were grown in tissue culture until passage 3 (P3) was reached and re-suspended as 5 × 10^3^ or 5 × 10^4^ cells per mL in StemMACS media (Miltenyi Biotec). In triplicate, 200 μL of cell suspension (either 10^3^ or 10^4^ cells, of either periosteum or BM MSCs) was pipetted onto PCL3%-E (4 cm^2^), sterilised under UV light for 1 h in a class II fume hood, and left to attach for 4 h, at 37 °C, 5% CO_2_, and grown in vitro. At timepoints over 4 weeks, samples were taken, washed in PBS, fixed in 3.75% (vol. %) formaldehyde overnight, and imaged using confocal microscopy (20× magnification, Leica Confocal Microscope, DM6 CS), staining protocols in Appendix A.

### 4.7. Modified Transwell Barrier Function Assay

A modified Transwell migration assay was developed to assess the barrier function of PCL3%-E in comparison to PCL3%-S. Sterilised membrane samples (1.3 cm^2^ pieces) were contained within autoclaved MINUSHEET^®^ tissue carriers and tension rings (Minucell) [72]. Constructs were placed into a 24-well plate floating on top of 500 μL of media and serum starved (suspended in FCS free DMEM media) P3 periosteum MSCs were seeded on top of the membranes at a density of 5 × 10^3^ cells per cm^2^. Time points were fixed and transferred facing upwards to a histology slide to be imaged with confocal microscopy.

### 4.8. Statistical Analysis

Following a D’Agostino & Pearson normality test, datasets were analysed using one-way ANOVA, with a Tukey’s multiple comparisons post hoc test (parametric data), a Kruskal–Wallis test (non-parametric data) or an unpaired *t*-test. Statistical significance was taken as *p* < 0.05 and data were plotted as mean ± standard deviation.

## 5. Conclusions

Upon histological investigation, IM was significantly thicker with smaller diameter blood vessels than periosteum, however a similar vascularity was seen through tissue blood vessel density, confirming the immature status of IM in comparison to periosteum. A commercially scalable manufacturing technique, FSE, was selected to generate a PCL based fibrous barrier membrane that can be customised post manufacture to create increased diameter pores. PCL3%-E does not only absorb and release PL at comparable rates to a commercially available membrane, but also robustly supports the proliferation and alignment of periosteum and BM MSCs whilst maintaining a barrier to cell migration. This membrane shows clinical promise for the application of surgical repair of CSBDs, where new barrier membranes to contain grafting material act as a carrier for growth factors or autologous sources of MSCs, whilst being able to support periosteal regrowth during bone regeneration are needed.

## Figures and Tables

**Figure 1 ijms-21-05233-f001:**
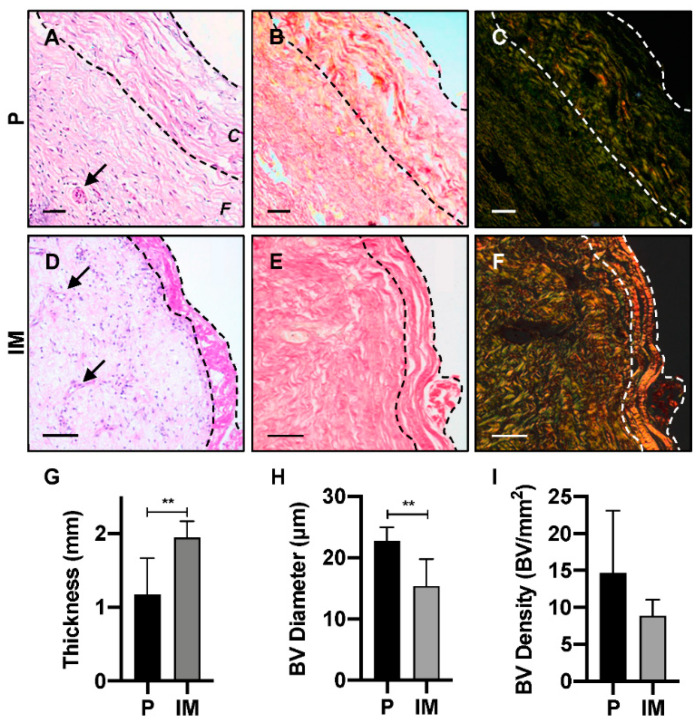
Histological analysis of periosteum (P) and IM samples. (**A**–**F**) Analysis of tissue architecture and cellular location, using haematoxylin & eosin (left column) and Picro Sirius Red staining under light microscopy (middle column) and polarised microscopy (right column). (**G**) Tissue thickness, (**H**) blood vessel diameter and (**I**) blood vessel density were quantified. Scale bar represents 50 μm. Dotted lines—outline cambium layer, arrow—blood vessel (BV), *C*—cambium layer, *F*—fibrous layer. Unpaired t-test, ** *p* < 0.004.

**Figure 2 ijms-21-05233-f002:**
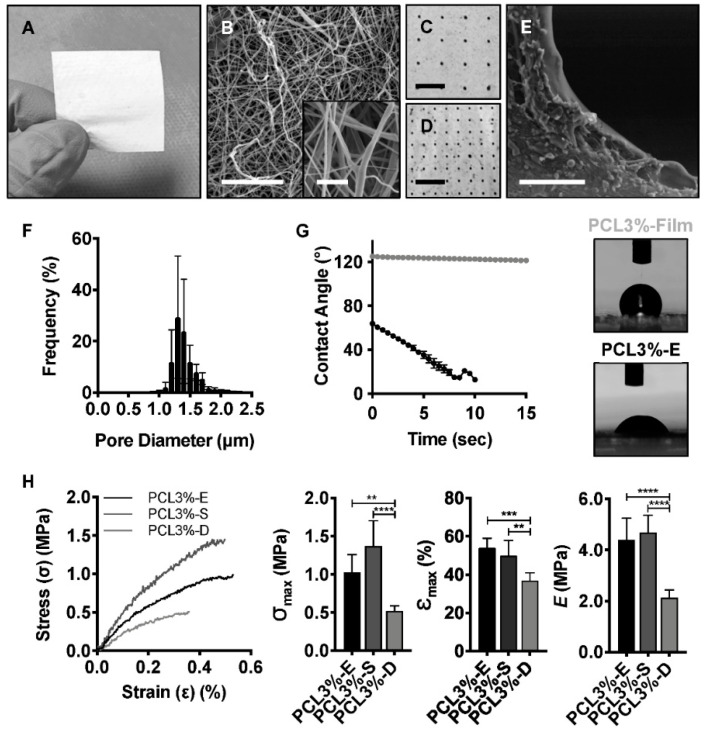
PCL3%-E characterisation and post-manufacture modification. (**A**) Macro image of PCL3%-E, showing ease of handling. (**B**) SEM images of randomly orientated nanoscale fibres. Scale bars: 25 μm; insert 2.5 μm. Macro images of laser cut membranes, (**C**) PCL3%-S, with sparse inter-pore distance (4 mm) and (**D**) PCL3%-D, with dense inter-pore distance (2 mm). Scale bars: 0.5 cm. (**E**) SEM image of laser cut pore edge. Scale bar: 100 μm. (**F**) PCL3%-E pore size frequency distribution. (**G**) Water contact angle measurements and image of pore-free cast films and PCL3%-E. (**H**) Stress-strain curves for electrospun membranes and ultimate stress, strain at break, and elastic modulus reported for each experimental group (from left to right). One-way ANOVA with Tukey’s multiple comparisons post hoc test, ** *p* < 0.005 *** *p* < 0.0005 **** *p* < 0.0001.

**Figure 3 ijms-21-05233-f003:**
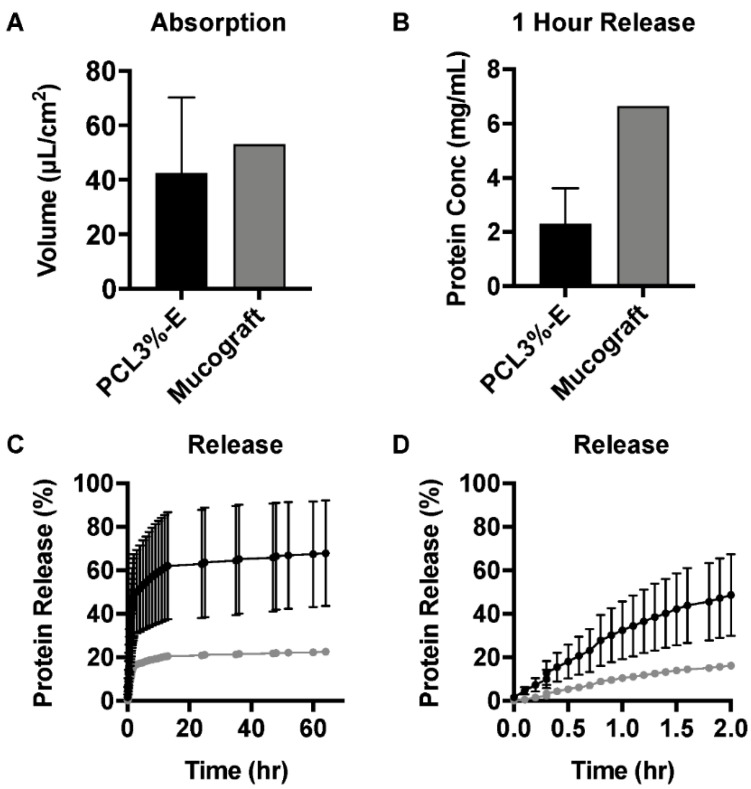
Protein absorption and release profiles from PCL3%-E in comparison to Mucograft^®^. (**A**) Absorption capacity and (**B**) protein release in the first hour. (**C**) The cumulative protein released as a percentage of the initial absorbed volume shown over 60 h and (**D**) the first 2 h. Black—PCL3%-E (*n* = 2), grey—Mucograft^®^ (*n* = 1).

**Figure 4 ijms-21-05233-f004:**
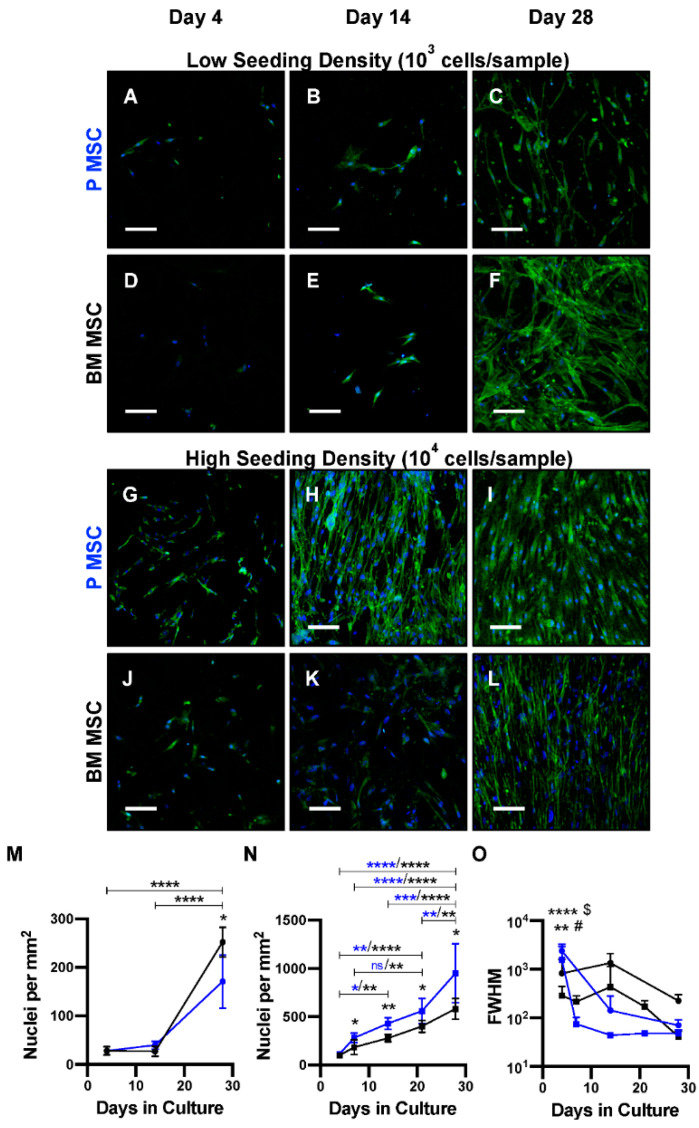
Attachment and proliferation of periosteum (P) and bone marrow (BM) MSCs onto PCL3%-E over a 28-day cell culture. Low (**A**–**F**) and high (**G**–**L**) seeding densities were used with both P MSCs **(A**–**C**,**G**–**I**) and BM MSCs (**D**–**F**,**J**–**L**). Confocal imaging following DAPI (blue, nuclei) and Phalloidin (green, actin filaments) staining. Nuclei count was quantified (*n* = 5), showing proliferation over time for (**M**) low and (**N**) high seeding densities. (**O**) Cellular alignment shown via measuring nuclei angle (*n* = 5). Gaussian distribution plotted and full width at half maximum (FWHM) measurements calculated as a measurement of data spread. P MSC—blue/blue stars, BM MSC—black/black stars, 10^3^ cells/sample—circle, 10^4^ cells/sample—square. Unpaired t-test (P vs. BM MSCs) and one-way ANOVA with Tukey’s multiple comparisons post hoc test (time point comparison) carried out, * *p* < 0.05, ** *p* < 0.005, *** *p* < 0.001, **** *p* < 0.0001. Significance at day 4 vs all other time points, ^$^ P MSC (10^3^), ^#^ P MSC (10^4^). Scale bar—100 μm.

**Figure 5 ijms-21-05233-f005:**
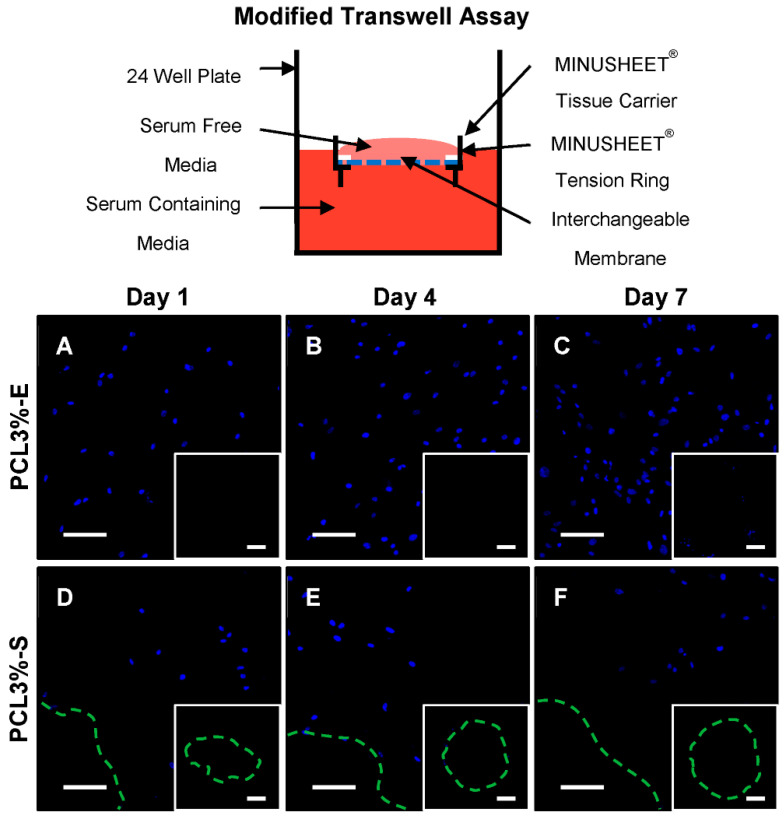
PCL3%-E barrier function assessment. Schematic of assay set up, MINUSHEET^®^ tissue carrier, containing (**A**–**C**) PCL3%-E or (**D**–**F**) PCL3%-S. The bottom of the well (dark red) had serum containing media and serum starved MSCs (10^3^ cells/sample) (pink), were added to the top of the MINUSHEET^®^ tissue carrier. (**A**–**F**) Constructs were imaged from the top (main image) and bottom (insert). Green dashed line corresponds to the pore edge on laser cut samples. Scale bar: 100 μm.

**Figure 6 ijms-21-05233-f006:**
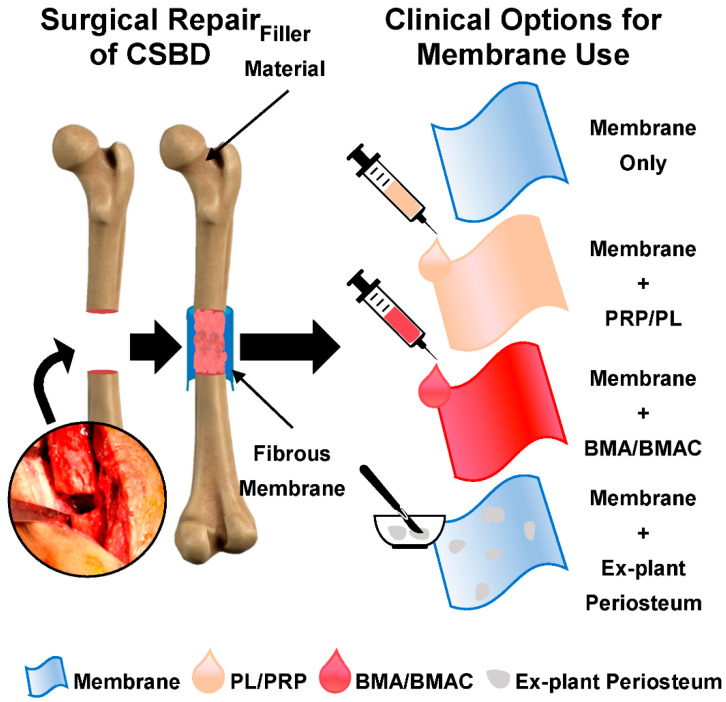
Schematic for surgical repair of CSBDs, with use of barrier membrane to wrap around the defect site, containing grafting material. The membrane can be used on its own or in combination with various blood products and MSC sources, dependent on the clinical need. PRP—platelet rich plasma PL—platelet lysate, BMA—bone marrow aspirate, BMA—BMA concentrate.

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
