# Peer review of "Induced Periosteum-Mimicking Membrane with Cell Barrier and Multipotential Stromal Cell (MSC) Homing Functionalities"

_ijms, 2020, doi:10.3390/ijms21155233_

Round 1

Reviewer 1 Report

I congratulate the authors for this study on guided bone regeneration, a classic topic that has been previously exposed by multiple authors using different materials to act as containment and guide membranes. Nevertheless I have some concerns about the specific solution that the authors propose. 

Are the physical characteristics, thickness, porosity, distance interpores, and the mechanical properties of the membrane, optimal for its handling, implantation and durable performance?

How are they changing while de biodegradation of the membrane?

How is the biodegradation process? How long does it last?

Why have you compare PL loading/release capability of the PCL3%-E membrane with that of Mucograft, being the use of one and the other so different?

To what do you attribute the different behaviour of bony of periostal MSCs depending on the seeding density?

I thank you for your good job!  

Author Response

Comment 1: I congratulate the authors for this study on guided bone regeneration, a classic topic that has been previously exposed by multiple authors using different materials to act as containment and guide membranes. Nevertheless I have some concerns about the specific solution that the authors propose. 

Response 1: We  thank the reviewer for taking the time to review and for their positive appraisal of our work and the relevant points raised below.

Comment 2: Are the physical characteristics, thickness, porosity, distance interpores, and the mechanical properties of the membrane, optimal for its handling, implantation and durable performance?

Response 2: The membrane is sufficiently robust to be handled, supported by tensile strength values of 1 MPa, reported in Figure 2 (Line 137), which we acknowledge is important to ensure clinical application. The dimensional properties are indicated in Figure 2 and as part of section 3.2. We found the membrane able to be folded to increase thickness 4-fold (line 131-132) and mean pore size was 1.4 μm (line 122), optimal for the membranes function to prevent cellular infiltration whilst allowing for nutrient absorbance and transport. We recognise that due to testing being in vitro only, future in vivo studies would be required to test the membranes’ functionality for handling, implantation and degradation over time.

Comment 3: How are they changing while de biodegradation of the membrane?

Response 3: We have not investigated the degradability of the membrane as this investigation would have been difficult to integrate in a single manuscript. It can be speculated that the thickness (and pore size) of the membrane would decrease (and increase) according to a bulk hydrolytic degradation mechanism. Following an initial induction time where uptake of water by the membrane occurs, subsequent cleavage of the fibre-forming PCL chains will take place prior to significant microscopic breakdown of the pore wall, mass loss and reduction of tensile properties. We have modified the discussion to highlight this point in more detail, see lines 256-258.

Comment 4: How is the biodegradation process? How long does it last?

Response 4: The biodegradation of PCL is well documented, however, in vitro and in vivo degradation tests for our membrane are needed. As an aliphatic polyester, PCL has been widely reported to degrade via a bulk erosion hydrolytic degradation mechanism, whereby the degradation time (≤ 2 years) is typically affected by polymer’s molecular weight, degree of crystallinity, presence of end groups and solution pH (Woodard and Grunlan, 2018). This detail has now been highlighted in the discussion (lines 256-258 and 328-329).

Comment 5: Why have you compare PL loading/release capability of the PCL3%-E membrane with that of Mucograft, being the use of one and the other so different?

Response 5: We agree with the reviewer that Mucograft is used for a different purpose, namely maxiofacial applications. We used it as a positive control in our absorbance/release experiments as it has a very high absorbance capacity (Nica et al., 2020) and was freely-available. In response we have modified text in Results and Discussion (lines 157-158 and 266-271).

Comment 6: To what do you attribute the different behaviour of bony of periosteal MSCs depending on the seeding density?

Response 6: We believe this is multifactorial – inherent differences between BM and P MSCs in vivo can be seen through P MSCs growing along aligned collagen fibres within periosteum (Foolen et al., 2008; Shi et al., 2014), whereas BM MSCs attach onto the surface of bone within the BM (Ilas et al., 2019). This is suggestive that the material was similar to periosteum ECM, thus allowing periosteum MSCs to naturally align. We thank the reviewer for their valuable comment and have added our thoughts on this in Discussion (lines 293-294).

Comment 7: I thank you for your good job!  

Response 7: We appreciate the reviewer’s positive comments on our work.

Reviewer 2 Report

It was my pleasure to review the article titled Induced periosteum-mimicking membrane with cell barrier and multipotential stromal cell (MSC) homing functionalities by Owston et al.which describes the development of a membrane produced by a free surface electrospinning technique for the treatment of critical size bone defects. The article is well written, well designed, and with good results showing a potential use in clinical applications.

There are some points that need to be clarified and described more in detail.

1) 3.3 PL Absorption and Release Profile of PCL3%-E 

Could it be possible to have more data about the absorption and release study? PCL3%-E  (n=2), Mucograft® (n=1) With this number of samples studied is difficult to get a real conclusion.

2) 3.4 Cellular Interactions with PCL3%-E 

It would have been important and interesting to study the osteoblast's interaction with the membrane as well as the MSCs derived from either human BM or periosteum to mimic the clinical situation where a bone graft will interact with this membrane.

3) Figure 6 Clinical Options for Membrane Use 

It is not clear nor described if in the clinical application the membrane will need to be associated or not with bone grafts or biomaterials.

4) 2.2 Histology Analysis of Periosteum and Induced Membrane 

Could you please describe more in detail the quantification of tissue thickness, surface area, blood vessel diameter, and frequency method used?

5) Conclusions 

This in vitro study shows the potential use of this membrane in clinical application. This study needs to be validated with an orthotopic animal study.

It would be important to add for example that an orthotopic animal study will confirm the potential use of this membrane for bone regeneration.

Author Response

It was my pleasure to review the article titled Induced periosteum-mimicking membrane with cell barrier and multipotential stromal cell (MSC) homing functionalities by Owston et al. which describes the development of a membrane produced by a free surface electrospinning technique for the treatment of critical size bone defects. The article is well written, well designed, and with good results showing a potential use in clinical applications.

Response: We thank the reviewer for their comments and raising the points below.

There are some points that need to be clarified and described more in detail.

1) 3.3 PL Absorption and Release Profile of PCL3%-E 

Could it be possible to have more data about the absorption and release study? PCL3%-E  (n=2), Mucograft® (n=1) With this number of samples studied is difficult to get a real conclusion.

Response 1: We completely agree with the reviewer and we would have liked to increase the number of replicates for this experiment, but are currently unable to carry out experimental work in the lab due to COVID, this is also likely to be for the foreseeable future. Since we used the same commercial platelet lysate product in these experiments, n numbers reflect the number of materials samples used in these tests. Mucograft being a commercial product with a well-defined absorbance/release capacity (Nica et al., 2020) is expected to show limited batch variation. The fact that despite the observed batch variability in PCL3%-E we detected similar absorption volumes but considerably higher protein release from Mucograft after 1 hour (Figure 3A-B) is encouraging and we have modified the text accordingly (lines 157-158 and 266-268) and the need for more replicates in these experiments is also highlighted in discussion (lines 269-271).

2) 3.4 Cellular Interactions with PCL3%-E 

It would have been important and interesting to study the osteoblast's interaction with the membrane as well as the MSCs derived from either human BM or periosteum to mimic the clinical situation where a bone graft will interact with this membrane.

Response 2: Indeed, future work would include osteoinduction of MSCs when seeded onto the material (lines 306-309), however, as a lab we work with primary human MSCs, rather than osteoblastic cell lines, thus the first steps were to investigate MSC attachment and growth. We have included these considerations, with supporting references (Wang et al., 2011; Shi et al., 2014), in the Discussion (lines 306-309).

3) Figure 6 Clinical Options for Membrane Use 

It is not clear nor described if in the clinical application the membrane will need to be associated or not with bone grafts or biomaterials.

Response 3: We have clarified in the text that the material would be in close contact with bone graft and/or biomaterials as the ‘filler material’ (lines 274-277). An arrow to ‘filler material’ is also added to Figure 6.

4) 2.2 Histology Analysis of Periosteum and Induced Membrane 

Could you please describe more in detail the quantification of tissue thickness, surface area, blood vessel diameter, and frequency method used?

Response 4: We have included more detail on the histology quantifications in the main manuscript methods, see section 4.2, lines 343-348.

5) Conclusions 

This in vitro study shows the potential use of this membrane in clinical application. This study needs to be validated with an orthotopic animal study.

It would be important to add for example that an orthotopic animal study will confirm the potential use of this membrane for bone regeneration.

Response 5: Clarification in the text that in vivo studies need to be carried out in order to not only quantify in vivo membrane degradation, but also contributions to bone regeneration (lines 328-329). We have also added reference to a recent relevant in vivo study (Lammens et al., 2020).

Round 2

Reviewer 1 Report

The revision of text made by the authors improves the quality of the report of the study. Some concerns have been resolved and the questions posed well answered. Changes done in a good way of better understanding.

Thank you very much.

Even the existence of some weak points of this study, I think it could be taken into account for publication.